# How the External Visual Noise Affects Motion Direction Discrimination in Autism Spectrum Disorder

**DOI:** 10.3390/bs12040113

**Published:** 2022-04-18

**Authors:** Nadejda Bocheva, Ivan Hristov, Simeon Stefanov, Tsvetalin Totev, Svetla Nikolaeva Staykova, Milena Slavcheva Mihaylova

**Affiliations:** 1Department of Sensory Neurobiology, Institute of Neurobiology, Bulgarian Academy of Sciences, 1113 Sofia, Bulgaria; ivanhristovnb@gmail.com (I.H.); simeonstefanov@abv.bg (S.S.); cwetalin@abv.bg (T.T.); milenski_vis@abv.bg (M.S.M.); 2Department of Psychiatry and Medical Psychology, Sofia Medical University, 1431 Sofia, Bulgaria; svetla_petrova@gbg.bg

**Keywords:** visual perception, motion direction, Autism Spectrum Disorder, equivalent noise paradigm

## Abstract

Along with social, cognitive, and behavior deficiencies, peculiarities in sensory processing, including an atypical global motion processing, have been reported in Autism Spectrum Disorder (ASD). The question about the enhanced motion pooling in ASD is still debatable. The aim of the present study was to compare global motion integration in ASD using a low-density display and the equivalent noise (EN) approach. Fifty-seven children and adolescents with ASD or with typical development (TD) had to determine the average direction of movement of 30 Laplacian-of-Gaussian micro-patterns. They moved in directions determined by a normal distribution with a standard deviation of 2°, 5°, 10°, 15°, 25°, and 35°, corresponding to the added external noise. The data obtained showed that the ASD group has much larger individual differences in motion direction thresholds on external noise effect than the TD group. Applying the equivalent noise paradigm, we found that the global motion direction discrimination thresholds were more elevated in ASD than in controls at all noise levels. These results suggest that ASD individuals have a poor ability to integrate the local motion information in low-density displays.

## 1. Introduction

Autism refers to a broad range of conditions characterized by impaired social interaction, verbal and nonverbal communication, and restricted or circumscribed interests with stereotyped behaviors [1]. Together with social, cognitive, and behavioral deficiencies, peculiarities in sensory processing [1], including visual perception, were found [2,3,4].

Investigations conducted during the last two decades have shed light on and have raised questions about the underlying neuronal mechanisms participating in the processing of visual motion information in Autism spectrum disorder (ASD). The atypical global motion processing in ASD has been explained by different theories, such as the increased vulnerability of the dorsal [5] or magnocellular [6] pathways. In addition, Gepner and Féron [7] suggested that motion processing in ASD may be affected by poor integration of information over space and time. Other theories suggest that more general impairments could affect abilities to process motion information. For instance, enhanced processing of local details [8] or a reduced ability to integrate information [9] could affect the integration of motion information in ASD. Other researchers [10] have found an increased sensitivity to noise rather than diminished integration ability. Their results showed no difference in the performance of the two groups in the no-noise condition and impaired performance of the ASD group when adding stimulus noise to visual motion through a cloud of dots. To explain these findings, the authors assumed an increased sensitivity to sensory noise and diminished usage of prior knowledge in ASD.

The coherent motion paradigm [11,12] is one of the most widely used experimental procedures to investigate psychophysically the motion processing properties. It evaluates the ability to integrate local motion signals into a global motion pattern [13]. In the coherent motion experiments, the stimuli are a population of coherently moving dots (signal) mixed up with different amounts of randomly moving dots (noise). The perception threshold is measured by estimating the minimum signal-to-noise ratio or the lowest proportion of signal dots required to detect coherent motion or determine its direction [14,15,16]. Several studies found elevated thresholds in ASD compared to TD using coherent motion tasks [17,18,19]. The abnormal increase in coherence motion thresholds could be due to impaired estimation of local motion stimuli caused by an increased internal noise [20], by insufficient evidence accumulation [21], or poor integration of local direction signals into global motion perception [14]. A recent meta-analysis of coherent and biological motion [16] shows that the small global motion processing deficit is independent of the task, age, and intellectual abilities of the studied groups. The authors suggested that the reduced performance in ASD is predominantly related to their inability to segregate the signal from noise dots. The presentation time, dot number, and speed do not moderate this effect significantly.

Robertson et al. [22] considered the coherent motion perception into two processing stages: 1/the detection of momentary local motion signals in the environment, occurring in V1 and MT/V5; 2/the spatiotemporal integration of these signals into a global percept or decision-variable, taking place presumably in the parietal cortex. In the coherent motion tasks, the pooling of multiple local motion signals with different directional variance would ameliorate direction estimates and thus combat external and internal noise on local motion signals [20]. The equivalent noise (EN) paradigm is a methodology that allows separating the local and global contributors to the task performance. It has been proved as a highly efficient approach for investigating mechanisms underlying the effect of various cognitive, developmental, and disease states of the sensory systems. Developed by engineers and formulated as an equivalent input noise method, its basic principle, when applied in perceptual neuroscience, comprises the idea of two sources of variability–external noise coming from the output signal and intrinsic internal neuronal noise. When the external noise is much less than the internal noise, the variability, and therefore the signal-to-noise ratio, are determined mainly by the internal noise. Vice versa, when the external noise is much greater than the intrinsic noise, the variability and signal-to-noise ratio are determined primarily by the external noise [23].

Using a novel application of the EN paradigm, Dakin and his co-workers evaluated the local and global limits of motion integration [14]. The authors showed that both local and global limits in direction integration depended on the number of moving elements.

Manning et al. [24] used the EN paradigm to investigate motion integration performance in ASD and TD children. No difference between the two groups was obtained in the no-noise condition. Despite the similar internal noise levels in both groups, the ASD subjects showed enhanced motion pooling in high external noise conditions compared to controls. However, the enhanced motion integration in ASD did not reach significance in a later study with a new sample of participants [25]. Only combining the results from the two studies provided sufficient evidence for a better motion integration in ASD than in TD. Other studies using a variant of the EN paradigm also do not show group differences in the behavioral performance [26], though differences were observed in the N2 component amplitude in the visual evoked responses.

In summary, the previous studies using the EN paradigm are equivocal regarding the enhanced motion pooling in ASD. Manning et al. [24] explained the advantage of the ASD group in motion integration by an inappropriately narrow range of direction pooling of the TD group. If this were the case, using a lower density stimulus would increase the ASD advantage. The present study aimed to evaluate the abilities of children and adolescents with ASD in comparison to the TD controls to integrate global motion information using a low-density display and the EN paradigm. In addition, we decided to include more noise level conditions than were studied previously [24].

## 2. Materials and Methods

### 2.1. Participants

Fifty-seven children and adolescents participated in the study: 26 in the ASD group (10 girls, 16 boys; M = 11.6 years, SD = 2.33) and 31 in the TD group (7 girls, 24 boys; M = 11.6 years, SD = 2.33). The ASD subjects were recruited from the Sofia Center for Social Rehabilitation and Integration– autism spectrum priority, the Regional Center for Support of the Inclusive Education Process-Sofia-city, and via community organizations, parental associations, and professionals (psychologists, speech therapists, child psychiatrists, etc.). The TD participants were recruited from different regular schools with permission and support of the Regional Department of Education–Sofia city.

Because ASD is a heterogeneous neurodevelopmental disorder and commonly co-occurs with other psychiatric conditions or neurological disorders, we examined the presence of comorbidity in the ASD group. Parents of all participants filled out a questionnaire. They had a brief semi-structured interview about the child’s developmental, medical history (neurological, psychiatric disorder, head trauma, current medication, and sensory impairment that could interfere with the performance of tasks), educational and behavioral concerns, etc. There are no data on comorbidity in children in the ASD group. None of the children in the TD group had a history of developmental problems or any mental or neurological diagnosis.

Wechsler Intelligence Scale for Children–Fourth Bulgarian Edition (WISC–IV BG, 2015; Wechsler, 2003) was administered to both groups. It resulted in the Verbal Comprehension Index (VCI), Perceptual Reasoning Index (PRI), Working Memory Index (WMI), Processing Speed Index (PSI), and Full-Scale IQ (FSIQ) (see Table 1).

All children in the ASD group had already been diagnosed with a pervasive developmental disorder (including Autism, Asperger’s syndrome, and ASD) according to ICD-10 (International Statistical Classification of Diseases and Related Health Problems 10th Revision, 1990) criteria. They had a written report of the autism diagnostic assessment from a child psychiatrist. For the purpose of the study, the diagnosis was confirmed by an experienced clinical psychologist using the Autism Diagnostic Interview–Revised (ADI–R) [27,28], reviewing their most recent developmental and medical reports and data from the developmental questionnaire. The ADI-R is a standardized semi-structured interview conducted with the parents about their child’s developmental history and autism symptoms that yield ratings for qualitative abnormalities in reciprocal social interaction (Score A), language and communication (Score B), restricted, repetitive, and stereotyped patterns of behaviors (Score C), and abnormality of development (Score D). It provides a diagnostic algorithm according to the most commonly used diagnostic frameworks-ICD-10, DSM-IV, and DSM-5. It comprises 93 items, 42 of which can be ranked into the following four scores with the respective cut-off values for diagnostic purposes: Score A- 10; Score B- verbal 8; Score C- 3; and Score D- 1. As all the participants in the experimental cohort scored above the cut-off level in each of the three domains of ADI-R and exhibited some abnormality in at least one area by the age of 36 months, they were classified as patients with autism (see Table 2).

There were no significant differences in the age between the groups t(55) = 0.16, *p* = 0.87 and sex t(55) = 1.30, *p* = 0.20. As expected, an independent-samples t-test confirmed that the two groups did differ in levels of intellectual functioning and WISC score: FSIQ t(55) = 5.08, *p* < 0.001, VCI t(55) = 6.43, *p* < 0.001, WMI t(55) = 4.77, *p* < 0.001., PRI t(55) = 1.94, *p* = 0.02 and PSI t(55) = 4.23, *p* < 0.001.

All participants had normal or corrected-to-normal near and far visual acuity, measured by Rosenbaum Pocket Vision Screener and Tumbling “E” Test, respectively, at 35.6 cm and 3 m. All had 1200” stereo acuity measured by Lang’s stereo test and normal contrast sensitivity measured by Hamilton-Veale Contrast Sensitivity Test. Five of the participants in each group dominantly used the left hand.

### 2.2. Stimuli and Procedure

The stimuli were generated by custom software and presented on an EIZO CS230 23″ monitor with a vertical refresh rate of 60 Hz and a screen resolution of 1920 × 1080 pixels. The mean display luminance was 18 cd/m^2^. The stimulation field had a size of 40° × 22.5° (ratio 16/9). The monitor’s default settings and calibration were checked and controlled by X-Rite i1 Eye-One Monitor Calibrator. Custom software written in C + + was used to generate the stimuli by an OpenGL video card and control the experiment.

The stimuli consisted of 30 moving Laplacian-of-Gaussian micro-patterns, i.e., the second derivative of the Gaussian (Figure 1). The average brightness of the micro-patterns was equal to the background to avoid edge and brightness effects. The dot density was 0.095 dots/deg^2^.

The micro-patterns (Figure 2) moved in a circular aperture with a diameter of 20° at a speed of 4 deg/s in different directions. If any element reached the edges of the aperture, it reappeared at a random position inside it, keeping its previous motion direction. The individual elements of the stimuli had an SD of 4.85′.

They moved for 833 ms in directions determined by a normal distribution with a SD of 2°, 5°, 10°, 15°, 25°, 35°, corresponding to the added external noise level. Stimuli with different noise levels were presented in separate blocks. The first trial in the block was triggered by the participant who pressed any button on a controller. The participant’s response to the previous trial triggered the subsequent trial. The new trial started after 800 ms of an intertrial interval with the appearance of a blank grey screen of mean luminance with a fixation dot in the center along with a warning beep. A moving stimulus replaced the blank screen after a fore-period that varied between 500 and 1000 ms.

The participant’s task was to determine whether the average motion direction was to the left or right of the vertical and press the left key when the perceived motion direction was to the left and the right key when it was to the right. The observers’ responses were collected through the color-coded keys on a joystick controller. The keys on the left side of the response controller were colored in red and the keys on the right side in green to correspond to the colored lines in the uppermost part of the screen. A custom device processed the responses and transmitted them to a computer.

A two-down, one-up double staircase procedure combined with the two-alternative forced-choice method was used to measure the motion direction discrimination thresholds. The mean stimulus direction changes by 10% after each reversal.

The experiments were divided into two sessions of 4–6 blocks of trials, performed on different days to reduce tiredness. Self-timed breaks were given to the participants between the blocks. Training trials were performed before the first session until the participants proved they fully understood the task and could correctly perform it.

The participants sat in a darkened room without direct sunlight. The viewing was binocular. The distance to the screen was 70 cm, and it was periodically verified.

After the researcher thoroughly explained the procedure, parents provided informed written consent, and each participant gave oral consent. The decision concerning participation in the study was entirely voluntary and could be withdrawn at any time. All the participants received a voucher as a reward for their participation. The experimental procedure was in accordance with the ethical standards of the Declaration of Helsinki and its later amendments and was approved by the Ethics Committee of the Institute of Neurobiology, Bulgarian Academy of Sciences (reference EC-INB/2018.40). All participants understood the task and were cooperative, as proved by their performance in training trials.

## 3. Results

The results show more considerable individual differences in external noise effect on motion direction thresholds in ASD than in TD. The individual global motion direction discrimination thresholds of the children from both groups are presented in Figure 3 as a function of the noise level. The variability between the participants increases with the elevation of noise level for both groups, but the variance among ASD children is much higher, especially for the noisiest conditions.

The motion direction thresholds are regarded as a measure of response variability. To specify the noise effect on performance, the data were fitted by a variance-summation model in which the response variability depended on two independent noise sources – internal and external. Dakin and co-workers [14] related the internal noise to the precision of local velocity estimates and the efficiency in using the available stimulus information in the presence of external noise to the number of samples pooled by the visual system in global motion direction estimation.

A two parameters function was used to represent the relation between the motion direction thresholds and the external noise:(1)$$\sigma_{\mathrm{thresh}}=\sqrt{\frac{\sigma_{\mathrm{int}}^{2}+\sigma_{\mathrm{ext}}^{2}}{N}}$$

In Equation (1), σ_thresh_ is the motion direction threshold, σ_int_ is the internal noise, σ_ext_ is the external noise, and N corresponds to the number of samples integrated into global motion direction estimation.

Nonlinear mixed model analysis was performed to determine if the parameters for the two groups differ. To account for the data heterogeneity, we used a random-effects model. The analysis showed that the two groups have similar sensitivity to the local motion direction. The parameter that reflects the number of samples used for global direction determination (N) is significantly larger for the TD than the ASD group implying that the TD group explores the global motion information more effectively. The values of the fitted parameters for the two groups are: σ_int_ = 7.07 [4.48–9.66]; N = 1.56 [1.32–1.80] for the ASD and σ_int_ = 7.19 [3.99–10.38]; N = 2.84 [2.30–3.39]–for the TD group. These results suggest that the two groups do not differ significantly in the level of internal noise, but the TD group explores more efficiently the stimulus information by integrating more samples in motion direction estimation.

Figure 4 illustrates the dependence of the mean motion direction thresholds for the two groups of participants on the external noise levels and the fitted functions evaluated by Equation (1). For both groups, the thresholds gradually increase with the external noise. They are higher for ASD individuals than for TD subjects, and the difference between them increases with the heightening of the noise level.

A potential reason for the within-subject variability observed in the thresholds might be the differences in participant characteristics. To test this possibility, we estimated the correlation between the parameters representing the sensitivity to local motion direction (σ_int_) and the number of integrated motion directions (N), and the intellectual abilities of the participants. Table 3 represents the correlation between individual estimates of these parameters for each group and the different scales of the WISC–IV: FSIQ, PSI, WMI, PRI, and VCI.

The results show that the global motion direction discrimination in the ASD group depends very little on the characteristics measured by the WISC-IV scale. Only the PRI correlates significantly with the ability to integrate local motion signals. The performance of the TD group is more related to the scores from the WISC-IV. The precision of local motion direction estimation correlates with PSI, whereas the integration ability correlates with the FSIQ and VCI.

No significant correlation was observed between the participants’ age and the additive internal noise (r = −0.21; *p* = 0.30 for ASD; r = −0.18; *p* = 0.33 for the TD group) or integrated samples (r = 0.14; *p* = 0.50; r= −0.01; *p* = 0.96 for the ASD and TD groups, respectively).

## 4. Discussion

The results obtained show substantial dissimilarities in the global motion discrimination performance between the ASD and TD groups. Our findings showed much larger differences in external noise effect on motion direction thresholds in the ASD than in the TD group, resulting in a higher variance in the ASD group. The global motion direction discrimination thresholds were more elevated in ASD compared to controls at all noise levels. Moreover, the thresholds gradually increased with the increasing external noise in both groups, more apparent for those with autism. These results imply lower sensitivity to mean motion direction in ASD than TD, especially in high external noise. Applying the equivalent noise paradigm, we found non-significant differences in local motion direction sensitivity, representing the additive internal noise. This result agrees with the findings of Manning et al. [24] in the no-noise condition. However, contrary to their results [24], which suggested motion direction enhancement in the ASD group to the TD, our results imply inferior performance at high noise levels. As discussed in our previous study [29], the discrepant results of different studies on visual perception in ASD may be a result not only of the difference in the experimental procedure but also of the sample choice and the individual characteristics of the participants. The sample groups in our study and in Manning et al. [24] have quite similar age characteristics. However, in our study, the two groups are equated in age but not in IQ scores in WISC-IV. We tried to include a representative sample for the heterogeneity of the disorder, and participants with ASD were not excluded based on their lower cognitive functioning. Brown et al. [30] underlined the significance of including vision research data from individuals with IQ below 80 to characterize the spectrum accurately. The authors found that 80% of the published papers included high IQ samples, which are not representative of the disorder.

Interestingly, in [25], the no-noise threshold measuring the sensitivity to local motion direction correlated negatively with the Performance and Verbal IQ scores. In our study, it is the number of pooled samples that correlated with the PRI score of the ASD group. This index is composed of subtests measuring perceptual and fluid reasoning, spatial processing, and visual-motor integration. PRI score depends on the ability to cope with problem tasks, on building concepts, planning and implementing their solution, relying on their visual-motor and visual-spatial skills. More IQ scores (FSIQ, PSI, VCI) correlate with the task performance parameters in our study for the participants with typical development.

There are several differences in the stimulation between the present work and the study of Manning and colleagues [24], such as the aperture size, speed, dot density, and stimulus presentation time. To explain the enhanced motion pooling in ASD in their study, Manning et al. [24] supposed that the typical children integrated the direction information over an inappropriately narrow range. This explanation seems incorrect as the dot density in our experiments was much lower than in [24]. If this were the case, one would expect much lower pooling for the TD group, whereas we obtained better performance. One possible reason for the narrowing of the integration range for the TD children in the high-noise condition in [24] could be the presence of a fixation point during the stimulus presentation. Since the TD group kept a more stable fixation, their observation area is more limited. This will not affect the motion discrimination in the no-noise condition, as even a single element could determine the direction.

Another potential explanation for our results might be the broader range of external noise levels. As discussed in [20], the increase in external noise also increases the multiplicative noise. At higher noise levels, the mean motion direction used in our study naturally deviated more from the vertical direction. Hence, the task might turn to coarse motion direction discrimination. For this case, [20] showed a correlation between autistic traits and multiplicative noise. In [24], the mean motion direction was fixed in the direction integration task at high noise but the standard deviation varied. In contrast, in our study, the mean motion direction depended on subject’s response, but the standard deviation was fixed. As discussed in [16], fluctuations in a decision variable could explain the multiplicative noise. Dissimilarities in later processing stages were also observed in [31] in motion integration tasks. The authors showed differences between the ASD and the TD groups in the response-locked activity typically related to decision-making processes.

The possibility that at high noise levels, the multiplicative noise is related to the fluctuations in the decision variable might explain the correlations observed between the number of samples integrated by the TD group and the IQ scores – a positive correlation with FSIQ and VCI scores, and with PRI – for the ASD group. The VCI is composed of subtests measuring verbal abilities utilizing, reasoning, comprehension, and conceptualization. The composition of this index involves less crystallized knowledge and more of an emphasis on reasoning and comprehension. In this sense, we could also think about the impact of the different coping strategies that the participants use. The PSI is connected to focusing attention and distinguishing and consistently arranging visual information. In our study, it correlates with the estimated additive internal noise of the TD group. This result suggests higher sensitivity (lower internal noise) for individuals that are better able to focus attention. Processing visual material quickly is an ability that participants from the ASD group perform poorly compared to children from the TD group [32]. Processing speed indicates the rapidity with which children can mentally process simple or routine information without making errors. Performance on this task may be influenced by visual discrimination and visual-motor coordination. Usually, learning involves a combination of routine information processing (such as reading) and complex information processing (such as reasoning); relative weakness in the speed of processing routine information may make the task of comprehending new information more time-consuming and difficult for children. This relative weakness in simple visual scanning and tracking may leave less time and mental energy for the complex task of understanding new information.

Despite the two-dimensional coherent motion of elements on the screen, the difference in individual elements’ directional movement generates an illusion of various depth planes’ motion. This phenomenon is related to the extraction of the common motion component ascribed to the background motion [33]. The residual motion of the individual elements is interpreted as speed differences and leads to perceived depth from motion parallax. A previous study on motion direction discrimination [32] shows that different depth planes obtained by spatially band-pass elements’ speed variability increase the pooling of motion information. The integration of motion information across apparent depth planes might be one of the reasons for decreased ability to integrate local motion signals among ASD children.

## 5. Conclusions

The study’s findings show significant differences in the global motion discrimination performance between the ASD and TD groups. The observed differences are not due to a greater inaccuracy in local direction estimation but to lower efficiency in global motion integration. The data obtained suggest that ASD individuals have poor ability to integrate the local motion information in low-density displays due to inferior ability to integrate motion information across apparent depth planes or disturbance of connectivity between low-level visual areas. Our results confirm the suggestion of [20] that even small changes in the experimental conditions could significantly influence the performance in the presence of noise.

## Figures and Tables

**Figure 1 behavsci-12-00113-f001:**
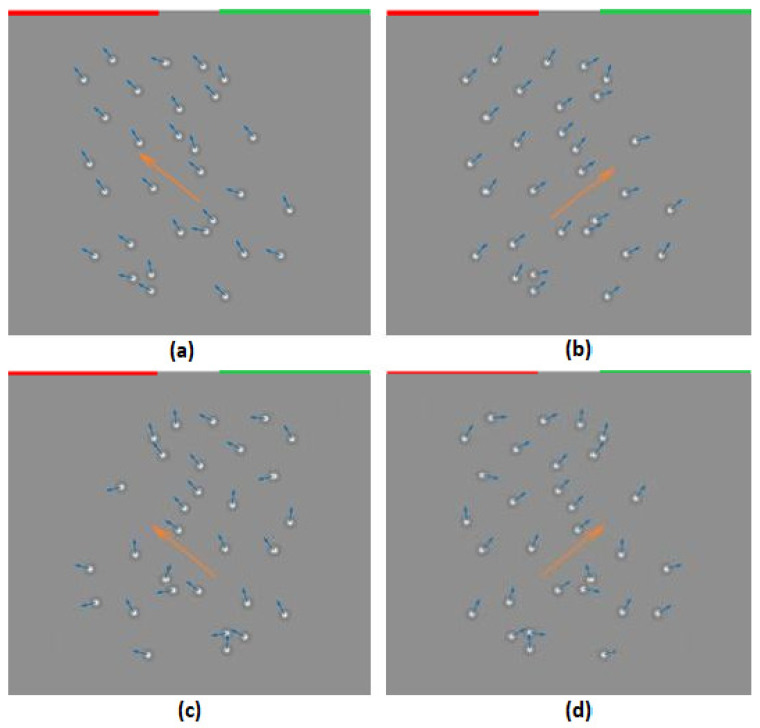
Schematic stimulus representation. Mean dot movement to the right (**a**,**c**) and to the left (**b**,**d**) of the vertical. The added external noise is 5° (**a**,**b**) or 15° (**c**,**d**).

**Figure 2 behavsci-12-00113-f002:**
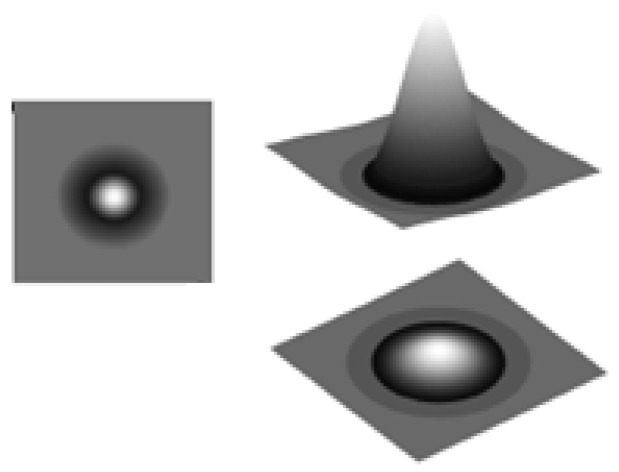
Example of the micro-patterns used in the experiments.

**Figure 3 behavsci-12-00113-f003:**
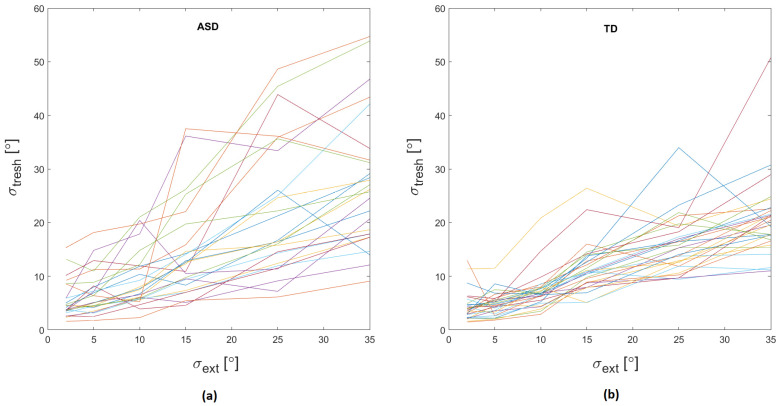
The dependence of global motion direction discrimination threshold on the noise level for the two groups of observers: (**a**) ASD group; (**b**) TD group. The lines represent the dependence of thresholds on the noise level for each participant.

**Figure 4 behavsci-12-00113-f004:**
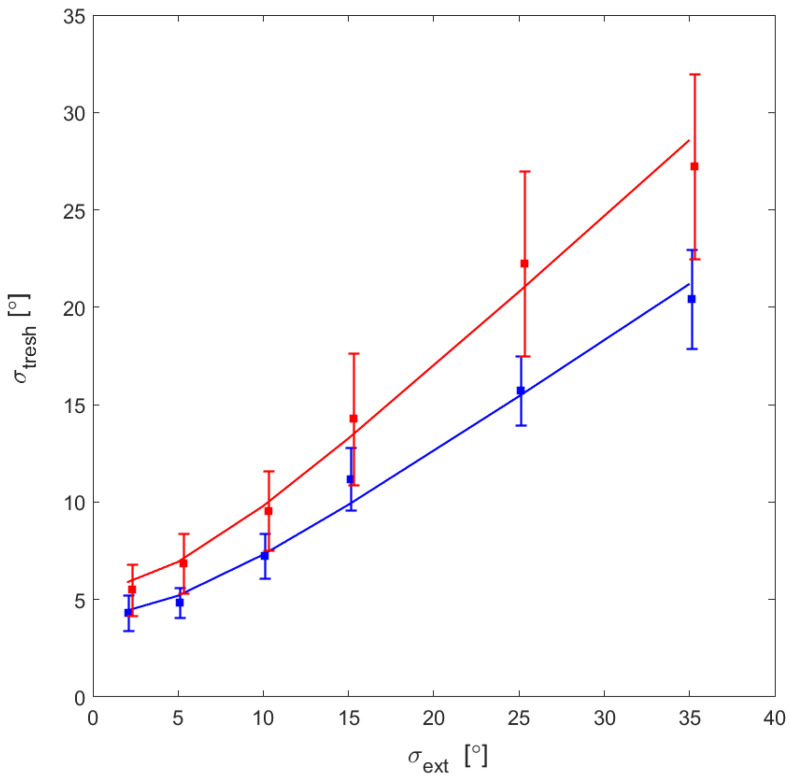
The mean discrimination thresholds for the two groups and their 95% confidence intervals. The solid lines show the functions fitted to the data by Equation (1). The data for the ASD are in red, and for the TD group in blue.

**Table 1 behavsci-12-00113-t001:** Results of Wechsler Intelligence Scale for Children–Fourth Bulgarian Edition.

	ASD Group (N = 26)	TD Group (N = 31)
N (Female/Male)	26 (10/16)	31 (7/24)
Age Mean ± SD [range] in years	11.7 ± 2.26 [8–16]	11.6 ± 2.33 [8–16]
WISC-IV (Mean ± SD [range])		
VCI	80.15 ± 18.43 [45–124]	105.58 ± 11.04 [85–142]
PRI	88.42 ± 22.55 [50–136]	100.09 ± 13.36 [76–129]
WMI	85.54 ± 17.81 [59–123]	103.87 ± 10.87 [77–123]
PSI	83.88 ± 16.60 [55–139]	99.94 ± 11.98 [78–124]
FSIQ	82.73 ± 17.19 [59–122]	103.00 ± 12.88 [80–141]

**Table 2 behavsci-12-00113-t002:** Results of ADI-R standardized semi-structured interview.

	ASD Group (N = 26)
	ADI-R (Mean ± SD [range])
Score A Qualitative Abnormalities in Reciprocal Social Interaction	25.92 ± 4.39 [11–30]
Score B Qualitative Abnormalities in Communication	19.62 ± 4.06 [9–24]
Score C Restricted, Repetitive, and stereotyped behavior	7.27 ± 2.72 [2–12]
Score D Abnormality of Development Evident at or Before 36 Monts	4.34 ± 0.94 [2–5]

**Table 3 behavsci-12-00113-t003:** Correlation between the indexes of the Wechsler Intelligence Scale for Children and the parameters σ_int_ and N for ASD and TD groups. The values in the brackets represent 95% confidence intervals of calculated correlations. The correlation values statistically significant at *p* = 0.05 are marked with asterisks.

WISK-IV	ASD		TD	
	σ_int_	N	σ_int_	N
FSIQ	R = −0.33 [−0.61–0.03]	R = 0.36 [−0.04–0.65]	R = −0.37 [−0.64–0.02]	R = 0.34 * [0.09–0.67]
PSI	R = −0.37 [−0.66–0.02]	R = 0.38 [−0.01–0.67]	R = −0.45 * [−0.69–−0.12]	R = 0.23 [−0.13–0.54]
WMI	R = −0.22 [−0.56–0.18]	R = 0.26 [−0.14–0.59]	R = 0.04 [−0.32–0.39]	R = 0.26 [−0.11–0.56]
PRI	R = −0.29 [−0.61–0.11]	R = 0.39 * [0.0001–0.67]	R = −0.36 [−0.64–0.01]	R = 0.25 [−0.11–0.55]
VCI	R = −0.38 [−0.67–0.003]	R = 0.19 [−0.21–0.54]	R = −0.35 [−0.63–0.004]	R = 0.31 * [0.04–0.60]

## Data Availability

The data supporting reported results can be found at https://osf.io/hu7qc/, last accessed 17 April 2022.

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
