# Peer review of "How the External Visual Noise Affects Motion Direction Discrimination in Autism Spectrum Disorder"

_behavsci, 2022, doi:10.3390/bs12040113_

Round 1

Reviewer 1 Report

The peer-reviewed paper is logical and well-structured, thoughtful, and addresses a clear problem on which it takes a clear position. The following comments are minor:

1) Figure 3 is completely uncluttered. Is it not possible to choose a clearer treatment of the graph? What do the different colours mean?

2) I would consider presenting Eq. 1 - it is a completely trivial relation, yet no other relations for calculations are given in the paper.

3) Serious flaw - I would recommend to go through all the tables and consider what error they are working with. The number of decimal places should also be adjusted to this. E.g. in Table 1 . age 11.6 ± 2.33 (should be 11.6 ± 2.3 and even more so - does the sample really allow the sample to work with that precision? Isn't the age 11.5 ± 2.5?) This is the lens through which to view the entire paper, all the results, their bias and reliability.

4) I leave it to the discretion of the authors to make the discussion more comprehensive, using more relevant and up-to-date sources. As it stands, this is the minimum scope and structure for acceptance.

Otherwise, this is a very well done paper.

Author Response

Rev 1

The peer-reviewed paper is logical and well-structured, thoughtful, and addresses a clear problem on which it takes a clear position. The following comments are minor:

1) Figure 3 is completely uncluttered. Is it not possible to choose a clearer treatment of the graph? What do the different colours mean?

We changed the Figure's caption to make it easier to understand.

2) I would consider presenting Eq. 1 - it is a completely trivial relation, yet no other relations for calculations are given in the paper.

We added arguments for using the equation.
3) Serious flaw - I would recommend to go through all the tables and consider what error they are working with. The number of decimal places should also be adjusted to this. E.g. in Table 1 . age 11.6 ± 2.33 (should be 11.6 ± 2.3 and even more so - does the sample really allow the sample to work with that precision? Isn't the age 11.5 ± 2.5?) This is the lens through which to view the entire paper, all the results, their bias and reliability.

We corrected the precision of the data.

4) I leave it to the discretion of the authors to make the discussion more comprehensive, using more relevant and up-to-date sources. As it stands, this is the minimum scope and structure for acceptance.

All the sources included in the Discussion are intrinsically related to our data. Still, we added some more citations. 

Otherwise, this is a very well done paper.

Reviewer 2 Report

  • Summary: This manuscript presents an interesting investigation on how Autism children may perform differently from typical children in motion direction discrimination tests. 
  • Review comments: The motivation of the study is straightforward. The study was designed and carried out soundly. The results are interesting and fill the gap in our current knowledge. But the English writing needs significant improvements.
  • Specific comments: I have a few detailed comments on data presentation and the writing.
  • Page 1 in Abstract: (line 17) Please define "SD"
  • Page 1 in Introduction: (line 30-34) This sentence is too long to be understood. Try to use short sentences.
  • Page 1 in Introduction: (line 39) You cannot use citation index to replace the sentence's subject. 
  • Page 3 in 2.1 Participants: (line 120-121) It is surprising that the two groups' means and standard deviations are exactly the same. The authors should double-check the numbers.
  • Figure 3: The y-axis label was mistyped.
  • Figure 3: The figure legend label should explain what are the colored lines.
  • Page 7: (line 251) The authors should introduce this function and explain why to choose it.
  • Page 8: (line 278) "We wanted to know..." - based on what reasons/previously results/assumptions, you wanted to know this and performed the analysis? When writing a research paper, you do not just pile up the results. You explain them thoroughly with the underlying logic that led your study. 
  • Page 9 in Discussion: (line 310) Again, you cannot use a citation number to replace the study or authors you referred to. 
  • Page 9 in Discussion: (line 333) "If this were the case..." - should use "was" instead of "were".
  • The Introduction was poorly written in my opinion. It is too long in terms of overall length. The sentences are also too long - many were painful to read. I would recommend splitting these sentences into short, logically-interconnected sentences. 
  • The Results were poorly presented. Remember that you do not just show the numbers and plots. You explain them.

Author Response

Rev 2

  • Summary: This manuscript presents an interesting investigation on how Autism children may perform differently from typical children in motion direction discrimination tests.
  • Review comments: The motivation of the study is straightforward. The study was designed and carried out soundly. The results are interesting and fill the gap in our current knowledge. But the English writing needs significant improvements.

We used a program to the improvement of the spelling, grammar, and style.

  • Specific comments: I have a few detailed comments on data presentation and the writing.
  • Page 1 in Abstract: (line 17) Please define "SD"

“SD” was replaced by “standard deviation”.

  • Page 1 in Introduction: (line 30-34) This sentence is too long to be understood. Try to use short sentences.

The sentence was split.

  • Page 1 in Introduction: (line 39) You cannot use citation index to replace the sentence's subject.

“Gepner and Féron” was added before “[7]”.

  • Page 3 in 2.1 Participants: (line 120-121) It is surprising that the two groups' means and standard deviations are exactly the same. The authors should double-check the numbers.

We checked the values and corrected them.

  • Figure 3:The y-axis label was mistyped.

The figure was corrected.

  • Figure 3:The figure legend label should explain what are the colored lines.

We changed the figure’s caption.

  • Page 7: (line 251) The authors should introduce this function and explain why to choose it.

We added explanations about the choice of the function.

  • Page 8: (line 278) "We wanted to know..." - based on what reasons/previously results/assumptions, you wanted to know this and performed the analysis? When writing a research paper, you do not just pile up the results. You explain them thoroughly with the underlying logic that led your study.

We explained the reason for our interest in this relationship.

  • Page 9 in Discussion: (line 310) Again, you cannot use a citation number to replace the study or authors you referred to.

We added “Manning et al.”

  • Page 9 in Discussion: (line 333) "If this were the case..." - should use "was" instead of "were".

We replaced “were” with “was”.

  • The Introduction was poorly written in my opinion. It is too long in terms of overall length. The sentences are also too long - many were painful to read. I would recommend splitting these sentences into short, logically-interconnected sentences.

We split and shortened the sentences along with other language corrections. We also shorted the Introduction.

  • The Results were poorly presented. Remember that you do not just show the numbers and plots. You explain them.

We made changes in the Results section adding more interpretations of the data.

Round 2

Reviewer 2 Report

I am happy to see that the authors have made significant improvements to the manuscript. My previous comments are well addressed (thank you for your effort). The manuscript is presentable now.

My one last comment is: can you please make the four panels in Figure 1 higher resolution?  It is really hard for me to see the blue and orange arrows clearly.